# ATF4 Signaling in HIV-1 Infection: Viral Subversion of a Stress Response Transcription Factor

**DOI:** 10.3390/biology13030146

**Published:** 2024-02-26

**Authors:** Adrien Corne, Florine Adolphe, Jérôme Estaquier, Sébastien Gaumer, Jean-Marc Corsi

**Affiliations:** 1Laboratoire de Génétique et Biologie Cellulaire, Université Versailles-Saint-Quentin-en-Yvelines, Université Paris-Saclay, 78000 Versailles, France; acorne397@gmail.com (A.C.); florine.adolphe@uvsq.fr (F.A.); sebastien.gaumer@uvsq.fr (S.G.); 2CHU de Québec Research Center, Laval University, Quebec City, QC G1V 4G2, Canada; 3INSERM U1124, Université Paris Cité, 75006 Paris, France

**Keywords:** ISR, AIDS, immunity, mitochondria, ER stress, UPR

## Abstract

**Simple Summary:**

Activating transcription factor 4 (ATF4) is a transcription factor known to regulate genes associated with the sensing of cellular stress such as amino acid deprival, protein misfolding, growth arrest, and cell death. Despite its key role at the crossroads of immune and stress responses, the precise impact of ATF4 during viral infections remains unclear. Thus, ATF4 has a dual role in promoting cell survival or cell death, but also in limiting infection or participating in viral replication. In this review, we focus specifically on ATF4-mediated signaling during human immunodeficiency virus 1 (HIV-1) infection and examine the multifaceted role of ATF4 in this context. We also explore the potential role of its paralogue ATF5. This review also discusses the possible hijacking of ATF4 activation by viruses to modulate host immune responses. By highlighting the involvement of ATF4 in viral infections, this study gives valuable insights that could be used for further strategies to tackle viral infections, providing a deeper understanding of host–pathogen interactions.

**Abstract:**

Cellular integrated stress response (ISR), the mitochondrial unfolded protein response (UPRmt), and IFN signaling are associated with viral infections. Activating transcription factor 4 (ATF4) plays a pivotal role in these pathways and controls the expression of many genes involved in redox processes, amino acid metabolism, protein misfolding, autophagy, and apoptosis. The precise role of ATF4 during viral infection is unclear and depends on cell hosts, viral agents, and models. Furthermore, ATF4 signaling can be hijacked by pathogens to favor viral infection and replication. In this review, we summarize the ATF4-mediated signaling pathways in response to viral infections, focusing on human immunodeficiency virus 1 (HIV-1). We examine the consequences of ATF4 activation for HIV-1 replication and reactivation. The role of ATF4 in autophagy and apoptosis is explored as in the context of HIV-1 infection programmed cell deaths contribute to the depletion of CD4 T cells. Furthermore, ATF4 can also participate in the establishment of innate and adaptive immunity that is essential for the host to control viral infections. We finally discuss the putative role of the ATF4 paralogue, named ATF5, in HIV-1 infection. This review underlines the role of ATF4 at the crossroads of multiple processes reflecting host–pathogen interactions.

## 1. Introduction

One of the main regulators of cellular homeostasis is activating transcription factor 4 (ATF4). This bZIP domain transcription factor plays a crucial role in both the integrated stress response (ISR) and the mitochondrial stress response (MSR) [1,2,3]. In reaction to oxidative stress, misfolded protein accumulation, or nutrient deprivation, ATF4 triggers the expression of genes involved in cellular processes participating in the control of autophagy or cell death. Furthermore, the role of ATF4 in cellular responses to stress is intimately linked to its dimerization partners such as the transcription factor ATF5, an ATF4 paralogue found in mammals, which belongs to the ATF4 bZIP-domain transcription factor family [4,5]. ATF5 is also strongly involved in stress responses and can be a target of ATF4. Thus, ATF4 and ATF5 are key cellular factors that integrate various stress signals.

During viral infections, these cellular processes can be modulated by virus machinery. In this context, it has been shown that numerous viral infections modulate ATF4 activity, which contributes to or represses viral replication, depending on the virus (Table 1) and the targeted host cell. While ATF4 levels have been shown to be increased by human immunodeficiency virus-1 (HIV-1) infection and to promote viral replication [6,7,8,9,10,11], data concerning the implementation of this regulation remain scarce. Thus, apart from its transcription activity at the HIV-1 promoter, our understanding of the impact of its various cellular functions on the HIV-1 viral cycle remains patchy. Understanding the mechanisms of ATF4 activation in the context of HIV-1 infection also merits further exploration as ATF4 is envisaged as a latency-reversing agent (LRA) that could be capable of reactivating HIV-1, thereby presenting the chance to eliminate viral reservoirs that are a major obstacle to the eradication of the virus from the infected organism [10].

This article aims to provide an overview of the role of ATF4 in HIV-1 infection. We summarize models in which ATF4 activation has been reported during HIV-1 infection and focus on how HIV-1 can induce ATF4 activation. We then explore the consequences of ATF4 activation on host cells and viral cycle. Finally, we also consider that ATF5, the paralogue of ATF4 in mammals, could be a potent actor during viral infections.

## 2. HIV-1 Infection Regulates ATF4

### 2.1. ATF4 Is Up-Regulated during HIV-1 and SIV Infections

HIV-1 infection leads to the development of acquired immunodeficiency syndrome (AIDS) associated with the depletion of CD4^+^ T cells, mainly by apoptosis [65,66,67] which predicts further pathogenicity [68,69]. Of interest, the induction of ATF4 following HIV-1 infection has been observed in several in vitro models. It was first described in Jurkat T cells, in which ATF4 was up-regulated at both the transcript and protein levels 8 h post-infection and remained elevated 48 h later [6]. This increase in *ATF4* transcript levels was further observed in primary human CD4^+^ T cells at day 5 post-infection [8]. In a model of HIV-1 latency, *ATF4* transcript and protein levels are weakly detectable [10] but increased with viral reactivation [6,8,10]. In vivo, in monkeys infected with simian immunodeficiency virus (SIV), *ATF4* transcripts are also up-regulated in the gut mucosa. This increase occurred during the acute phase of SIV infection, i.e., 1 to 2 weeks post-infection, but are not observed during the chronic phase [8]. Viral proteins released in the microenvironment of infected cells such as Tat has been proposed to be sufficient for inducing *ATF4* gene expression through ER stress in non-infected cells [11]. Altogether, these observations show that *ATF4* expression is differentially regulated during both the early and chronic phases of viral replication, in infected but also non-infected cells, which are close to the former, suggesting that ATF4 induction is related to viral stress and plays a potential role in the establishment of latency.

### 2.2. How Can HIV-1 Regulate ATF4?

#### 2.2.1. HIV-1-Induced ISR/ATF4 Signaling

The ISR leads to a global translation blockade through the phosphorylation of eIF2α, which increases ATF4 translation [3]. Several kinases are responsible for eIF2α phosphorylation: double-stranded RNA-dependent protein kinase (PKR), PKR-like ER kinase (PERK), general control non-derepressible 2 (GCN2), heme-regulated eIF2α kinase (HRI), microtubule affinity-regulating kinase 2 (MARK2), and family with sequence similarity 69 member C (FAM69C) [3,70,71] (Figure 1). The role of eIF2α and its kinases during viral infection has been recently reviewed by Liu et al. [72]. We focus on HIV-1 studies providing evidence that eIF2α phosphorylation by ISR kinases can be associated with ATF4 induction, and that a direct control of these kinases by HIV-1 proteins may affect ATF4 activity.

Among the ISR kinases, PKR is the best characterized viral nucleic acid sensor [73,74]. It is activated mainly by double-stranded RNAs (dsRNA) [75,76,77,78]. PKR regulates HIV-1 infection as it can bind the hairpin structure within the transactivation-response region of the HIV-1 genome [79]. While PKR contributes to ATF4 translation in response to endoplasmic reticulum (ER) stress [80], paradoxically, to our knowledge, no study reports the induction of ATF4 in the context of HIV-1 infection through an endogenous PKR/eIF2α pathway. A reason that may explain this lack of evidence is the rapid inhibition of PKR by multiple mechanisms during HIV-1 replication [79] (Figure 1): PKR activation is inhibited by high levels of dsRNA and by the direct binding of cellular proteins, including TAR RNA-binding protein (TRBP) and adenosine deaminase acting on RNA (ADAR1) [81,82,83,84,85,86,87]. Moreover, the viral Tat protein may counteract PKR activation by preventing PKR autophosphorylation. Tat also serves as a competitive inhibitor limiting PKR binding to eIF2α thanks to a sequence homology between one of its motifs and eIF2α [88,89,90].

Cells experiencing ER stress due to nutrient deprivation or viral replication may lead to the activation of PERK. Like many other viruses, HIV-1 is known to induce protein misfolding and ER stress [91]. Campestrini et al. have previously reported on the induction of ER unfolded protein response (UPR) genes such as *PERK* and *ATF4*, following the treatment of Jurkat cells with the HIV-1 Tat protein [11]. Additionally, PERK activation has been observed in Tat-treated human brain microvascular endothelial cells (HBMECs) and in HIV-1-infected CD4^+^ T cell lines [92,93]. While PERK is one of the sensors of the UPR with IRE-1 and ATF6, the functional role of PERK in ATF4 induction in response to HIV-1 infection is not fully demonstrated.

GCN2 is activated in response to amino acid deprival and metabolic dysfunction (Figure 1). These alterations are reported in plasma of HIV-1-infected patients and in the gut of SIV-infected monkeys [8,94,95,96,97]. Moreover, GCN2 is activated in vitro by HIV-1 or SIV infection [8,98] suggesting that metabolic disorders observed in HIV-1-infected patients could also lead to GCN2 activation. GCN2 phosphorylates the HIV-1 integrase, which reduces viral integration [99]. This property is not limited to HIV-1 since various integrases from other retroviruses are also recognized as substrates by GCN2 [99]. The anti-viral activity of GCN2 can be overcome by the HIV-1 protease that can cleave GCN2 [100]. Jiang et al. have reported that HIV-1-induced ATF4 transcription would result from the activation of the GCN2/ATF4 pathway [8]. Serum deprivation triggers HIV-1 replication of CD4^+^ T cells, correlating with the GCN2-mediated activation of ATF4, which is recruited to the HIV-1 long terminal repeat (LTR) to facilitate viral transcription.

Both heme/iron depletion and arsenite-induced oxidative stress activate HRI [3]. Arsenite-induced HRI activation increases viral protein production and the infection rate of reovirus [101]. It can also regulate the level of the viral factor Zta that plays a role in the lytic cycle of EBV [102,103]. HRI has been shown to be able to activate ATF4, triggering the expression of genes involved in autophagy [104,105,106] and oxidative-stress response including the antioxidant heme oxygenase-1 (HO-1) gene [106]. Different groups have reported a role of HO-1 in regulating viral replication [107,108] including HIV-1′s [109]. HO-1 dysfunction has been associated with neuronal diseases in people living with HIV-1 [110]. For instance, despite a link between HO-1 and viral replication, to date, no study has addressed the role of an HRI pathway regulating HIV-1 infection and replication.

Recently, two other kinases able to phosphorylate eIF2α have been characterized, namely MARK2 and FAM69C [70,71]. Both kinases are induced by proteotoxic stress. FAM69C KO mice-derived microglia displayed an indirect increase in inflammation, and HIV-1 binds MARK2 to control motor adaptor function on viral cores [71,111]. However, no HIV-1-induced ATF4 activation depending on these kinases has been reported yet.

To conclude, PERK and GCN2 may lead to ATF4 activation after HIV-1 infection, and the direct interaction of HIV-1 components with GCN2 and MARK2 may in turn control their ability to activate ATF4. The potential control of ATF4 by the new eIF2α kinases needs further investigation during HIV-1 infection.

#### 2.2.2. Mitochondrial Stress Response, ATF4 and HIV-1

The contribution of HIV-1 infection to the induction of mitochondrial dysfunctions has been described earlier [112,113,114]. During HIV-1 infection, mitochondrial functions are compromised resulting in reduced oxidative phosphorylation (OXPHOS), ATP synthesis, gluconeogenesis, and β-oxidation. In addition to membrane depolarization and release of cytochrome C [115,116], HIV-1-induced alterations may also disrupt cellular homeostasis, increase oxidative stress, affect mitochondrial dynamics, and lead to the loss of mitochondrial DNA [115,117,118]. Mitochondrial dysfunctions are not only observed in CD4^+^ T cells, but also in myeloid cells such as neutrophils [119,120], monocytes [121], and CD8^+^ T cells [122], which are non-infected T cells. Therefore, indirect mechanisms may contribute to the alteration of mitochondrial functions during HIV-1 and SIV infections.

ATF4 is induced by several mitochondrial stress-like alterations affecting proteostasis, respiration, and mitochondrial membrane potential (MMP) loss [123,124,125]. A multi-omics study has suggested that ATF4 coordinates the mitochondrial stress response [125]. ATF4 induction was shown to depend on HRI in response to the respiratory chain and ATP synthesis disruptions [126,127]. Individual knockdown of each of the four original ISR kinases (i.e., HRI, PERK, PKR and GCN2) did not abolish the induction of ATF4 and its target genes. These results could suggest the role of either unknown or newly identified ISR kinases (as indicated above), or some redundancy.

In response to mitochondrial misfolded protein accumulation, ATF4 plays a significant role in cell homeostasis as a transcription factor of the canonical mitochondrial UPR (UPRmt) [124]. ATF4 is also associated with the transcription of genes encoding mitochondrial chaperonin and proteases [124]. For instance, Li et al. suggest that unfolded mitochondrial proteins would be degraded by lysosomes, leading to the increase in amino acids that would activate mTORC1 via lysosomal v-ATPase through a still-unknown mechanism during the UPRmt [128,129,130,131]. mTORC1 would then phosphorylate and activate ATF4, thereby triggering transcription of chaperone-encoding genes, and increasing the mitochondrial folding capacity [129].

ATF4 is also activated in response to alterations in mitochondrial dynamics. The deletion of optic atrophy protein 1 (OPA1), which is crucial for the fusion of inner membranes during mitochondrial fusion, and contributes to the release of apoptogenic factors [132,133], generates mitochondria-derived reactive oxygen species (ROS) and thus causes increased oxidative stress and death [117]. This process triggers ER stress and a PERK-dependent UPR, resulting in the transcriptional activation of *ATF4* and other genes. This response initiates a catabolic program contributing to muscle loss and systemic aging [134]. The knockdown of Drp1, a major effector of mitochondria fission [135], leads to eIF2α phosphorylation and ATF4 activation in the liver [136]. Although modulation of OPA1 was not directly associated with ATF4 activation, it has been show that the OMA1 protease, which cleaves the mitochondrial protein DELE1, but also OPA1 [137,138], leads to the release in the cytosol of the DELE1′s carboxy-terminal domain that oligomerizes with HRI (Figure 1) [126,127,139]. The authors propose that, in turn, HRI increases ATF4 protein levels [127]. Therefore, the effect on mitochondrial fission or fusion can be associated with an ATF4-mediated stress response. To date, ATF4 activation in the context of HIV-1 infection has not been reported to originate from mitochondrial stress. Nevertheless, this lack of data certainly stems from the fact that ATF4 is also involved in the response to other HIV-1-induced stresses that trigger ISR, like ER stress and amino acid depletion. It is thus challenging to determine the respective contribution of HIV-1-induced stress sources to ATF4 activation.

#### 2.2.3. The Viral Vpu Protein Stabilizes the ATF4 Protein

The HIV-1 viral protein U (Vpu) is an HIV-1 accessory protein that down-modulates CD4 and BST-2/tetherin. A cellular Skp, Cullin, F-Box (SCF) E3 ubiquitin ligase complex is recruited by Vpu to target CD4 for ubiquitination and proteasomal degradation [140]. This process involves the recruitment of the F-box β-transducin repeat-containing protein (βTrCP) [141,142,143,144]. Additionally, Vpu reduces BST-2/tetherin from the cell surface by preventing the trafficking of BST-2/tetherin to the plasma membrane from the trans Golgi network and/or the recycling endosome [145,146,147]. Furthermore, Vpu targets BST-2/tetherin for degradation, thus promoting viral progeny release and inhibiting NF-κB signaling [143,148].

The SCF-βTrCP E3 ubiquitin ligase complex has been reported to contribute in the degradation of ATF4 (Figure 1) [149]. Unlike its effect on CD4 and BST-2/tetherin, Vpu inhibits ATF4 βTrCP-mediated proteasomal degradation [150]. This apparent discrepancy in the effect of Vpu on βTrCP-dependent proteasomal degradation may be explained by the existence of two distinct paralogs of β-TrCP, βTrCP1/BTRC and βTrCP2/FBXW11 [151]. Recent work by Pickering et al. demonstrates that Vpu has contrasting effects on βTrCP1 and βTrCP2 and suggests that Vpu would induce proteasomal degradation mediated by βTrCP2 and inhibit βTrCP1-dependent protein degradation [152]. Thus, the contribution of viral proteins encoded by HIV-1 merits further investigation regarding the role of ATF4.

#### 2.2.4. HIV-1 Antiretroviral Drugs Induce ATF4 Signaling

HIV antiretroviral therapy (ART) has drastically altered the course of HIV-1 infection, resulting in a major decrease in morbidity and mortality. However, drug side effects have been reported earlier, leading to their progressive replacements and the development of new molecules. Thus, mitochondrial damage was initially reported following the use of reverse transcriptase inhibitors (RTIs) and protease inhibitors (PIs) [153,154,155,156,157]. In addition to mitochondrial stress, it has been shown that Nelfinavir (PI) triggers an ATF4 transcriptional response associated with liver metabolic alterations that have been reported in PLWH [158] and causes cell cytotoxicity against ovarian cancer cells [159,160]. In addition to Nelfinavir, it has been shown that Lopinavir (PI) also increases the level of ATF4 transcript in SQ20B and FaDu cancer cell lines [161] but in a model-dependent manner because no effect was observed on the level of ATF4 transcript in a model of trophoblast cell differentiation [162]. While Zidovudine (AZT, RTI), which induces mitochondrial stress, was recently reported to extend the lifespan of *C. elegans* depending on ATF4 activation [163], a reduction in ATF4 expression was reported [164] in long-term Tenofovir disoproxil fumarate (TDF)-treated individuals presenting a decrease in bone mineral density. Therefore, the role of ATF4 remains to be clarified regarding the use of RTIs. Interestingly, an HIV-1 integrase inhibitor (IN), namely MK-2048, was shown to selectively kill HTLV-1–infected cells by inducing the PERK/ATF4/CHOP pathway [165]. This molecule could be also of interest for people living with HIV-1 (PLWH), in which triggering the death of viral infected cells may reduce the extent of viral reservoirs. Thus, several ARTs may provide a beneficial effect not only by tackling HIV viral replication but also by stimulating ER stress via ATF4 and thus facilitating the death of infected cells.

## 3. ATF4 Role during HIV-1 Replication

### 3.1. ATF4 Positively Regulates HIV-1 Cycle

The induction of ATF4 during HIV-1 infection raises the possibility that ATF4 may play a role in viral replication (Table 2). Thus, it has been shown that the overexpression of *ATF4* promotes viral replication, whereas its silencing suppresses HIV-1 replication [6,7]. Using compounds that inhibit GCN2 or PERK or that lead to increases in ATF4 levels, it has been shown that the induction of the ISR/ATF4 pathway reactivates HIV-1 in models of HIV-1 latency [8,9,10]. Altogether, these data strongly suggest a role for ATF4 in regulating HIV-1 replication both during the acute infection and the exit from latency. However, given the nature of infected cells that include memory CD4 T cells and T follicular helper cells (Tfh) [166,167,168,169,170], which represent the main reservoirs in visceral tissues, further analyses should be performed to elucidate the role of ATF4 in primary T cell subsets.

### 3.2. How ATF4 Favorizes HIV-1 Replication

#### 3.2.1. ATF4 Binds to the HIV-1 LTR and Promotes Viral Gene Transcription

ATF4 can be a direct regulator of HIV-1 transcription. During HIV-1 infection, various cellular transcription factors including some bZIP domain proteins have been shown to bind the 5′ long terminal repeat (LTR)—that contains the transcriptional promoter of the viral genome of HIV-1—and regulates transcription of HIV-1 genes [171] (Figure 2).

ATF4 induces HIV-1’s gene transcription and regulates the Human T-cell Leukemia Virus (HTLV) promoter [172]. As an ATF/CREB family member, ATF4 binds to the C/EBP-ATF consensus sequence (TGACGT (C/A) (G/A)). Two C/EBP-ATF-binding sites were first identified in the LTR of HIV-1, but super-electrophoretic mobility shift assay (EMSAs) failed to show the binding of ATF4 to these regions [173,174]. A bioinformatics approach led by Jiang et al. identified additional potential C/EBP-ATF-binding sites in the LTR [8]. Caselli et al. reported that co-expression of Tat and ATF4 induced higher LTR transcription than the sole expression of Tat, suggesting a synergistic interaction between the two proteins [6]. ChIP experiments showed that ATF4 binds to the LTR sequence. This interaction is positively regulated by Forkhead box O 1 (FOXO1) inhibition and amino acids deprival [8,9]. Further works suggest that Tat may be unnecessary for ATF4-mediated HIV-1 promoter activation [6,7]. However, this effect has only been observed in HeLa cells, but not in Jurkat and 293 T cells [6].

ATF4 can also bind to sequences that differ from the C/EBP-ATF consensus sequence, depending on its dimerization partners (Figure 3) [1,175,176]. ATF4 can dimerize with Jun and Fos [177] that heterodimerize to form the AP-1 transcription factor [177]. The fact that the HIV-1 promoter presents an AP-1-binding site could suggest that a heterodimer of ATF4 with Jun or Fos can be formed to activate the LTR activity. However, the partners able to dimerize with ATF4 to control HIV-1 transcription are still unknown.

Five to twenty percent of HIV-1-infected patients are co-infected by the Hepatitis B virus (HBV). The X protein of HBV has been shown to regulate HIV-1 transcription by stimulating the binding of ATF4 to the LTR of HIV-1 [178]. ATF4 can also directly interact with the HTLV-1 transcriptional activator Tax protein and act as an adapter between Tax and the HTLV-1 promoter [172,179]. Therefore, co-infections may provide a permissive environment for ATF4 to activate the HIV-1 promoter.

#### 3.2.2. ATF4, HIV-1 and Apoptosis

It is well known that HIV-1 pathogenicity is associated with the depletion of CD4^+^ T cells leading to the development of AIDS. In this context, it has been shown that a form of programmed cell death, namely apoptosis, may contribute to the depletion of CD4^+^ T cells and be associated with pathogenicity [65,180,181]. Both in vitro and in vivo, HIV-1 and SIV infections mediate T cell death [67,182] through the intrinsic apoptotic pathway that is characterized by the expression of the pro-apoptotic proteins Bax, Bak, and Bim [183], leading to the release of apoptogenic mitochondrial factors and consequently to caspase activation. CD4^+^ T cell apoptosis also involves the extrinsic pathway in which Fas (CD95) is critical [184,185,186,187]. Thus, caspase inhibition prevents in vivo the depletion of CD4^+^ T cells and delays the progression to AIDS [66].

It has been shown that Tat up-regulates UPR mediators with an increase in the transcript levels of *ATF4*, *CHOP* and the pro-apoptotic BH3-only *BIM* (*BCL2L11*), thus leading to apoptosis [11]. Of interest, ATF4 mediates cell death through the activation of the bZIP transcription factor CHOP [3], which also induces the transcription of *BIM* and of the pro-apoptotic BH3-only *PUMA* (also known as *BBC3*) [188,189]. Additionally, the CHOP-ATF4 dimer up-regulates *ATF5*, *NOXA* (also known as *PMAIP1*), *APAF-1*, and *TXNIP*, which in turn amplify cell death [190,191,192]. ATF4 also promotes degradation of the caspase inhibitor X chromosome-linked inhibitor of apoptosis (XIAP) protein through the ubiquitin-proteasome system, which allows the activation of caspases [193]. Neill and Masson recently reviewed ATF4 target genes that may take part in various potent routes through which ATF4 may promote apoptosis [175] (Table 3). In a model of HIV-1 latency, inducing ISR/ATF4 by a prolonged pharmacological treatment, which is significantly associated with cell death [10], increases ATF4 and CHOP protein levels. Thus, several apoptotic genes have been found to be induced by ATF4 in CD4^+^ T cells during HIV-1 and SIV infections. However, the role of ATF4 in regulating the death of CD4^+^ T cells in the context of HIV-1-infection has been poorly explored.

Other genes have been identified [175] *that* could contribute to ATF4-mediated death of CD4^+^ T cells such as *Tumorous imaginal disc 1* (*TID1*), *Transmembrane Bax inhibitor-1 motif-containing protein 5* (*TMBIM5*), *G0/G1 switch gene 2* (*G0S2*), and *p53-binding protein 2* (*TP53BP2*).

TID1, also known as DnaJ homolog subfamily A member 3 (DNAJA3), which belongs to the DnaJ family of proteins is involved both in apoptosis and autophagy. It has been proposed that TID1 helps the oncosuppressor protein p53 to translocate to the mitochondria under hypoxic conditions, thus leading to a transcription-independent mitochondrial apoptotic pathway [218]. Given the role of p53-mediated death in productive HIV-1-infected cells [219,220], TID1 regulation of p53 merits to be further explored. Interestingly, *TID1* transcript levels are increased in T cells following HIV-1 infection, and a recent study shows that TID1 positively regulates HIV-1 replication [199,221].

TP53BP2, also known as apoptosis-stimulating of p53 protein 2 (ASPP2), can interact with p53, thereby enhancing its DNA binding and transactivation functions on the promoters of pro-apoptotic genes [222]. ATF4 regulates the *TP53BP2* gene in liver ischemia-reperfusion (I/R) injury and contributes to hepatocyte apoptosis [223]. In cells exposed to the HIV-1 glycoprotein gp120, the role of TP53BP2 would depend on stress intensity as exemplified by the use of varying doses of gp120 and could control both autophagy and apoptosis [216,224].

Another interesting but poorly described target of ATF4 is the TMBIM5 protein, also known as the growth hormone inducible transmembrane protein (GHITM, or MICS1). TMBIM5 localizes at the mitochondrial inner-membrane and is essential to maintain both mitochondrial structure and function [225,226]. This protein seems to ensure cell survival by allowing Ca^2+^ efflux from mitochondria and limiting mitochondrial hyperpolarization [227,228,229]. TMBIM5-knockout cells are more sensitive to apoptosis elicited by staurosporine or BH3 mimetic inhibitors of Bcl-2 and Bcl-xL [225,226]. Interestingly, global gene expression data show that *TMBIM5* transcript levels are down-regulated in infected monocytes and increased in astrocytes of HIV-1 HAND patients [214,215]. Given the limited data available, further investigations are needed to determine whether TMBIM5 plays a role in the induction of apoptosis downstream of ATF4 in response to HIV-1 infection.

Lastly, the *G0/G1 switch gene 2* (*G0S2*) that regulate lipid production [230] has also been found to be a transcriptional target of ATF4. Initially identified in cultured mononuclear cells in response to drug-induced cell cycle transition from the G0 to G1 phase [231,232,233], G0S2 displays opposite roles toward apoptosis. It prevents cells from ATP depletion, induces a cellular tolerance for hypoxic stress [234], and acts as a survival molecule in endothelial cells, protecting them from serum starvation- and hydrogen peroxide (H_2_O_2_)-induced apoptosis [235]. Interestingly, G0S2 is down-regulated in macrophages but induced in dendritic cells upon HIV-1 infection [200,201]. The biological significance of these observations has yet to be established.

Altogether, the identified ATF4 targets represent multiple new mechanisms to be explored by which ATF4 could regulate cell death or survival of HIV-1-infected cells.

#### 3.2.3. ATF4, HIV-1 and Autophagy

Autophagy is a pro-survival intracellular catabolic process that targets protein aggregates and damaged organelles in response to stress. ATF4 activates the transcription of genes involved in the initiation (*BECN1*), in the elongation of the phagophore and the maturation of the autophagosome (*WIPI1*, *ATG12*, *ATG5*, *ATG10*, *ATG16L1*, *ATG3*, *ATG7*, *MAP1lc3B*, *GABARAPL2*), and in the selective clearance of cargo (*p62*/*SQSTM1* and *NBR1*) [175,236]. Interestingly, ATF4 triggers the transcriptional activation of the autophagy genes directly as a homo or heterodimer with CHOP or indirectly through CHOP induction (Figure 3) [236]. The ratio between ATF4 and CHOP is decisive for the regulation of these genes, underpinning a fine-tuned control of the different stages of autophagy, the subtleties of which remain to be elucidated. ATF4 can also act upstream of autophagy to induce this process; ATF4 can increase the expression of the gene encoding Regulated in development and DNA damage response 1 (REDD1; also known as Ddit4), which suppresses mTOR complex 1 (mTORC1) activity, allowing autophagy induction upon ER stress and starvation [237,238,239].

HIV-1 modulates autophagy in a cell-type-dependent manner [240,241]. The Env protein found at the cell surface of infected cells induces an autophagy-dependent cell death of bystander CD4^+^ T cells [242]. Thus, ARF6, which promotes autophagosome biogenesis by facilitating endocytic uptake of the plasma membrane into autophagosome precursors, has been proposed to promote the fusion of HIV-1 envelope with the plasma membrane, therefore permitting the entry of HIV-1 in CD4^+^ T lymphocytes [243,244]. Other groups have reported that HIV-1 inhibits autophagy [245,246]. After virus entry, the Vpr protein decreases the amount of proteins like MAP1LC3 and BECN1 [247]. Furthermore, although autophagy is initiated in primary CD4^+^ T cells infected with HIV-1, this process aborts due to a lysosome destabilization by DRAM, a p53-inducible gene. This permeabilization of lysosomes and resulting cell death has been proposed as an altruistic self-defense mechanism limiting viral dissemination by the elimination of infected cells [220]. Despite the obvious link between ATF4 and autophagy, the role of ATF4 in autophagy in the context of HIV-1 infection remains poorly documented.

#### 3.2.4. Immune Response and ATF4 Activation during HIV-1 Infection


*ATF4, innate immunity and HIV-1*


Pattern recognition receptors (PRRs) are fundamental molecules for innate immune response recognizing pathogen-associated molecular patterns (PAMPs). PRRs bind to PAMPs and trigger cell signaling cascades. PRRs include Toll-like receptors (TLRs), Nucleotide-binding oligomerization domain (NOD)-like receptors (NLRs), RIG-I-like receptors (RLRs), and C-type lectin receptors (CLRs). For example, TLR7 and TLR8 can recognize single-stranded RNAs, which are present in the HIV-1 genome [248], leading to Interferon regulatory factor 3 and 7 (IRF3 and IRF7) activations and inducing robust expression of type I interferon (IFN) genes [249]. ATF4 inhibits IRF7 transcription but also its activation by inhibiting TANK-binding kinase 1 (TBK1) and IκB kinase (IKK)ε-mediated IRF7 phosphorylation [57]. To be activated, IRF7 also requires its ubiquitination by Tumor necrosis factor receptor-associated factor 6 (TRAF6) (Figure 3). GADD34 that binds to and inhibits TRAF6 activity is an ATF4 target [250]. Its transcript levels have been shown to be increased in HIV-1-infected cells in a p53-dependent manner [251]. Inhibition of TRAF6 enhanced HIV-1 replication in macrophages [252], whereas GADD34 reduced viral protein expression [253]. Therefore, the role of ATF4 and its targets merits to be further explored in innate immunity.

HIV-1 also up-regulates TLR2 levels and signaling both in vivo and in vitro [254]. This is of particular significance since HIV-1 provirus integration is increased in TLR2-bearing cells compared to cells that do not express TLR2 [255]. Although the role of ATF4 in this regulation of TLR2 has not been so far demonstrated, ER and oxidative stresses up-regulate *TLR2* mRNA levels in an ATF4-dependent manner [256,257]. A ChIP assay revealed a binding site for ATF4 in an intronic region of the *TLR2* gene [257]. Therefore, it can be hypothesized that ER stress induced by HIV-1 infections could activate ATF4 and regulate *TLR2* gene expression in infected cells. ATF4 also mediates TLR4-triggered cytokine production [258], and it has been shown that the level of TLR4 is increased upon HIV-1 infection [254]. Given the role of ATF4 in the regulation of TLRs, and that microbial antigens may activate innate cells, ATF4 activation could contribute to the induction of chronic inflammation that characterizes people living with HIV. Interestingly, transcriptomic analyses performed from the brain of HIV patients revealed an increase in ATF4 expression [259]. However, the work from Akay et al. indicated lower protein levels in HAND [260]. This apparent discrepancy highlights the necessity to clarify the role of ATF4 in HIV-1-induced inflammation.


*ATF4, cellular immunity and HIV-1*


Viruses have developed a diverse array of strategies to manipulate host cell metabolism and reorientate metabolic resources toward their benefit [118]. Current knowledge on T-cell metabolism in HIV-1 infection suggests that HIV-1 takes advantage of the glycolytic process in CD4^+^ T cells to infect them and to boost viral replication [261,262,263]. It has been shown that IL-7-mediated thymocytes survival through the increase in Glut1 protein levels at the cell surface, leading to glucose uptake and favoring viral infection [264]. Interestingly, ATF4 can influence a network of genes responsible for metabolic flux when activated by an extracellular oxidizing environment [265]. Thus, ATF4 up-regulates genes involved in glycolysis and promotes the anaplerotic flux through enhanced glutaminolysis, which plays a role in T cell growth in oxygen- or amino-acid-deprived environments [265]. Glutaminolysis fuels the tricarboxylic acid (TCA) cycle and oxidative phosphorylation (OXPHOS) in T-cell receptor-stimulated naïve CD4 subsets, as well as memory CD4 subsets [266]. This balance in the bioenergetics pathways is essential for T cell polarization [267]. ATF4 has also been implicated in regulating the differentiation of naïve T cells in T helper 17 cells (Th17). T cells from ATF4-deficient mice are characterized by a diminished Th1 differentiation and an elevation in factors promoting Th17 commutation, such as Interleukin-17 (IL-17) [265]. This association is further supported by the effect of the small molecule halofuginone (HF), which inhibits Th17 differentiation in both mice and humans and increases ATF4 protein levels [268]. Notably, adding amino acids in excess counteracts the inhibition of Th17 differentiation by HF, suggesting the involvement of the amino acid starvation response mediated by ATF4 in targeting Th17 differentiation. Interestingly, Th17 cells are highly vulnerable to HIV-1 infection [269], and this population represents a major viral reservoir under ART [270]. The loss of Th17 homeostasis contributes to disease progression in SIV-infected rhesus macaques associated with the loss of intestinal epithelial integrity, leading to microbial translocation [271,272,273,274]. This alteration in the balance of T cell subsets persists despite ART [275]. Given the role of ATF4 in regulating mitochondrial and reticulum stress-associated apoptosis, a contribution of ATF4 dysregulation in the alteration of the Th17 balance at the mucosal barrier cannot be excluded. Taken together, these results suggest that the induction of ATF4 observed in CD4^+^ T lymphocytes during HIV-1 infection [10] may contribute to the increased production of energy and amino acids required for viral replication.

## 4. ATF5 the Paralog of ATF4

ATF5 is a member of the bZIP transcription factor family, like CREB, Fos and NRF2 [276]. Based on its bZIP domain sequence, ATF5 has been classified into the ATF4 subgroup family (Figure 4). ATF5 also shares many functional features with ATF4, especially in stress cellular responses [277,278]. Interest in ATF5 has risen sharply in recent years, and its role in regulating cellular stress, cell survival, immunity, and cancer has recently been reviewed [279,280]. ATF5 is also involved in cell proliferation and differentiation of various cell types [276]. Thus, a growing number of studies implicate ATF4 and ATF5 in the regulation of the immune response and reveal how the dialogue between stress responses and immune pathways can be fundamental for the regulation of tissue homeostasis by the cell in response to infection [281]. Of particular interest, how ATF4/ATF5 and NF-κB coordinate their activity and their effects on viral replication, latency, and cell survival need further investigation.

Data concerning the involvement of ATF5 in the viral stress response and its possible role during infections remain sparse. During Epstein–Barr virus (EBV) infection, the viral protein Latent membrane protein 1 (LMP-1) positively regulates the ATF5 protein level via the NF-κB pathway [282]. In this context, ATF5 transcriptionally inhibits the *Signaling lymphocyte activation molecule-associated protein* (*SAP*) gene, promoting a Th1-like cytokine profile in T lymphocytes. Intriguingly, LMP-1 not only activates ATF5 but also positively controls PERK to induce phosphorylation of eIF2α, which up-regulates *ATF4* expression in B cells [43]. ATF4, in turn, transactivates *LMP-1*’s promoter in a positive feedback loop to enhance LMP-1 protein levels and activity. Thus, LMP-1 activates both ATF4 and ATF5, which seem to have different functions in the viral cycle.

ATF5 has also been shown to play a role in the replication of Herpes simplex virus type 1 (HSV-1). It promotes the replication of the virus by activating the transcription of viral genes, but this activity is counteracted by the cellular HSV-1 infection response repressive protein (HIRRP), through physical interaction [283]. Interestingly, an elevation in the eIF2α/ATF4 signaling is observed towards the end of HSV-1 replication, which leads to the completion of virion production and release [50]. Therefore, as in EBV infection, ATF4 and ATF5 act at different steps of the HSV-1 infection.

In glioma cells that are highly permissive to Human cytomegalovirus (HCMV) infection, the viral immediate-early protein of 86-kDa (IE86) interacts with and acetylates ATF5, thereby promoting cell survival [284]. Furthermore, the HCMV protein pUL38 is required to maintain the viability of infected fibroblasts by increasing phosphorylation of both PERK and eIF-2α and inducing robust accumulation of ATF4 protein [45]. Thus, both ATF4 and ATF5 contribute to the survival of infected cells.

Microarray and RNAseq analyses have shown that the levels of ATF5 transcripts increase in cells infected with viruses such as ZIKV or dengue [214]. ZIKV infection also increases ATF4 transcript and protein levels [38,39,285]. In the context of HIV-1 infection, ATF5 transcript levels are up-regulated in Th1Th17 vs. Th1 cells [286]. However, the roles of ATF4 and ATF5 are still poorly understood in these models. ATF5, like ATF4, can be activated by various stresses and involved in the ISR and UPRmt pathways. Therefore, whether ATF5 is activated in response to HIV-1 infection and regulated by HIV-1 proteins have to be determined.

## 5. Conclusions

In this review, we provide an overview of ATF4 functions during HIV-1 infection and explore the role of some of the poorly characterized ATF4 transcriptional targets that are involved in cell death, autophagy, metabolism, or in the immune response. This review thus proposes some molecular mechanisms that should be investigated for understanding the role of ATF4 in the context of HIV-1 but also by other viruses. Although most of the research on ATF4 is generally associated with the demonstration of protein and transcript levels modulation, it is also of interest to determine proteins relocation and ATF4-binding partners at promoter levels. It is also becoming well-established that post-translational modifications regulate ATF4 activity. This last point is less understood, and advances in our knowledge of the impact of post-translational modifications, particularly on the choice of ATF4 heterodimerization partners, should be decisive in understanding the control of ATF4 activity in the context of HIV-1 infection. Given its similarity with ATF5, it is crucial to decipher the roles of ATF5, depending on cell types and its fine regulations. Understanding the role of these transcriptional factors may pave the way to identifying new drug therapies.

## Figures and Tables

**Figure 1 biology-13-00146-f001:**
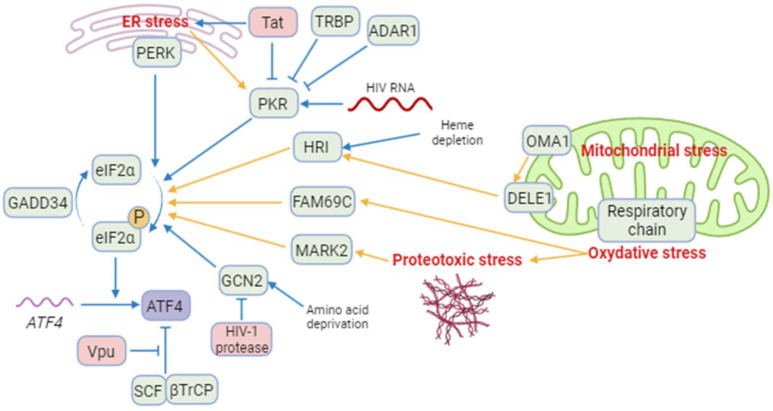
A model of regulation of ATF4 and integrated stress kinases during HIV-1 infection. Blue arrows correspond to interactions demonstrated by the literature in the context of HIV-1 infection. Yellow arrows correspond to reports in a different context. Besides ATF4, cellular proteins are indicated in green and HIV-1 proteins in red. ISR kinases can be activated by ER stress (PERK and PKR), HIV-1 RNA (PKR), heme depletion and cleavage of DELE1 by OMA1 after mitochondrial stress (HRI), amino acid deprival (GCN2), oxidative stress (FAM69C), and proteotoxic stress (MARK2). Several cellular and viral proteins modulate ISR kinases during HIV-1 infection, as TRBP and ADAR1 and the HIV-1 Tat protein preventing the phosphorylation of eIF2α and increasing ATF4 synthesis. The viral Vpu protein stabilizes ATF4 by opposing its ubiquitination by the SCF-βTrCP complex. Created with BioRender.com (accessed on 18 January 2024).

**Figure 2 biology-13-00146-f002:**
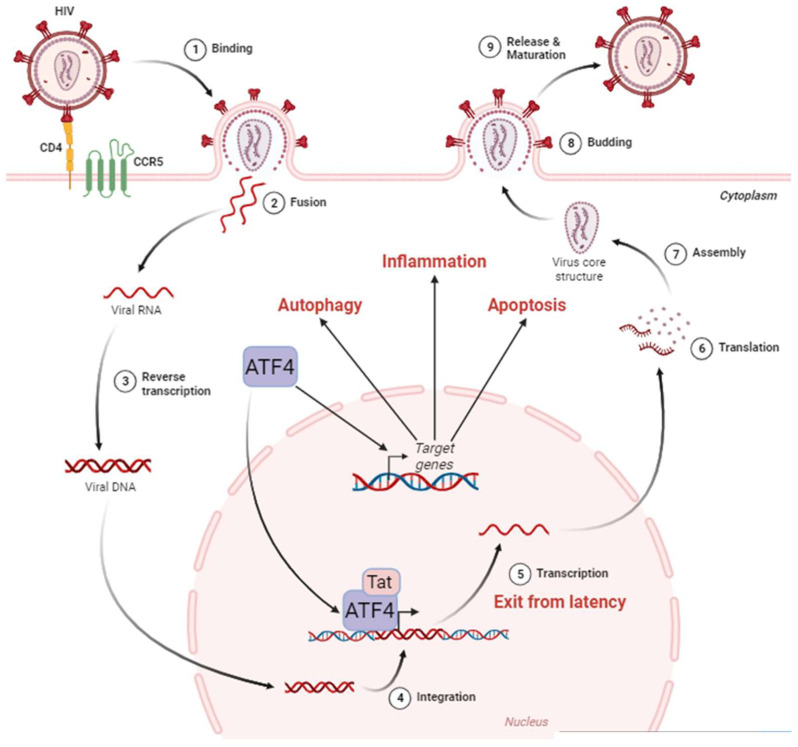
ATF4 in the HIV-1 cycle. The HIV-1 cycle begins with the attachment of the virus to the cell surface thanks to the CD4 and CCR5 receptors (1). This allows the fusion between the viral envelope and the host cell membrane (2). During this phase, the viral nucleocapsid enters the cell cytoplasm. Single-stranded RNA molecules are then retro-transcribed in the nucleus (3). The viral DNA then integrates into the host cell genome (4), where it can remain latent until reactivation. The exit from the latency is stimulated by ATF4, which, in cooperation with the viral protein Tat, activates the HIV-1 promoter located in the LTR region of the viral DNA leading to viral RNA (5). These HIV-1 messenger RNAs are exported from the nucleus to be translated (6) into proteins that assemble to form the internal structure of viral particles (7). Budding from the cell membrane (8) culminates with the release and maturation of viral particles in the extracellular environment (9). ATF4 is also capable of inducing the activation of target genes involved in cellular processes such as autophagy, inflammation, and apoptosis. Adapted from “Disease Mechanism–Infectious Diseases, HIV Replication Cycle”, by BioRender.com. Retrieved from https://app.biorender.com/biorender-templates (accessed on 18 January 2024).

**Figure 3 biology-13-00146-f003:**
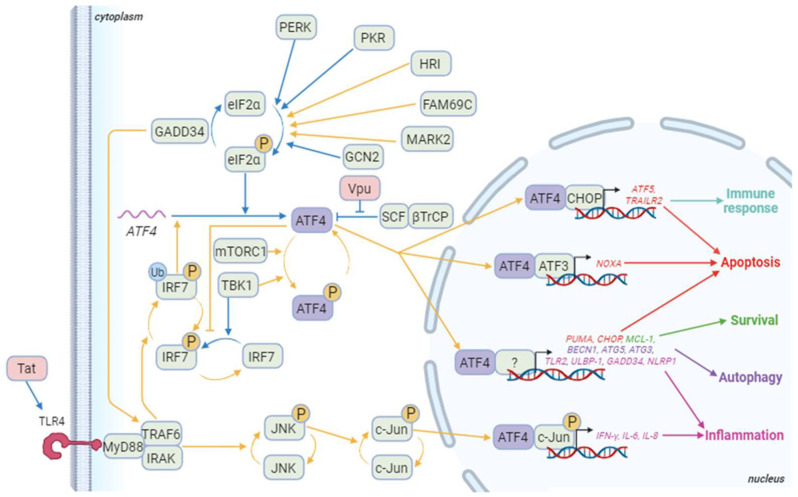
ATF4 signaling pathways during HIV-1 infection. Blue arrows correspond to interactions demonstrated in HIV-1 literature. Yellow arrows correspond to other contexts. Besides ATF4, cellular proteins appear in green and HIV-1 proteins in red. The genes in red are some of the pro-apoptotic genes induced by ATF4/ATF3 or ATF4/CHOP dimers and ATF4 dimerizing with a partner that remains to be determined. ATF5 also regulates the immune response at least by controlling immune cell differentiation. ATF4 also controls the expression of several pro-apoptotic genes including *TID1/DNAJA3*, *G0S2*, and *TP53BP2*. The genes in green are involved in ATF4-mediated survival as *TMBIM5*. Genes in purple are implicated in ATF4-induced autophagy like *WIPI1*, *ATG12*, *ATG10*, *ATG16L1*, *ATG7*, *MAP1lc3B*, *GABARAPl2*, *p62*/*SQSTM1*, *NBR1*, and *REDD1*. Finally, genes in pink are genes related to inflammation that are induced by ATF4 in a dimer with phosphorylated c-Jun or an undefined partner. Other genes activated by the phosphorylated ATF4/c-Jun dimer include *RANTES* and *sICAM-1*. Not shown in this diagram is IRF7 activation, which induces the production of type I and II interferons, and then activates the PKR/eIF2α/ATF4 pathway. Created with BioRender.com (accessed on the 18 January 2024).

**Figure 4 biology-13-00146-f004:**
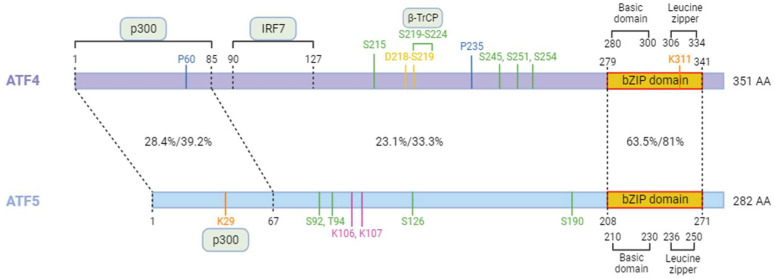
Comparison of ATF4 and ATF5 proteins. Percentages of identity and similarity between amino acid sequences of ATF4 and ATF5 are shown. Both proteins contain a bZIP domain comprising a basic region for DNA binding and a leucine zipper motif for dimerization. ATF4 interacts with the p300 protein thanks to its N-terminal region (1–85), allowing its acetylation by p300 on lysine 311. Hydroxylation on prolines 60 and 235 results in a decrease in ATF4 transcriptional activity. ATF4 has a DSGICMS motif recognized by the β-TrCP factor to induce its degradation by the proteasome, following its phosphorylation by β-TrCP on its serine 219. Phosphorylation of ATF4 by β-TrCP on serine 224 also participates in this regulation. ATF5 can be acetylated on its lysine 29 by p300. The first 21 amino acids of ATF5 are involved in stabilizing the protein in response to Interleukin-1β. The post-translational modifications of ATF4 and ATF5 are shown in the following color code: acetylation in orange, hydroxylation in blue, phosphorylation in green, ubiquitination in yellow, and SUMOylation in purple. Created with BioRender.com (accessed on 18 January 2024).

**Table 1 biology-13-00146-t001:** ATF4 regulation by viral infection and ATF4 role in viral replication.

Virus Family	Virus	Regulation of ATF4 (*)	Effect of ATF4 Regulation on Viral Replication (**)	Other Major Findings Related to ATF4 and Viral Infection.	Refs.
*Adenoviridae*	Adenovirus type 2 (AdV-2)	+ (t)	ND	ATF4 transcript is transiently increased before being down-regulated after the onset of the adenovirus early gene expression.	[12]
*Arteriviridae*	Porcine reproductive and respiratory syndrome virus (PRRSV)	+ (p)	[+]	ATF4 localizes to cytoplasmic viral replication complexes by the viral non-structural proteins nsp2/3.	[13]
*Asfaviridae*	African swine fever virus (ASFV)	− (p)	[+]	The viral protein DP71L inhibits the induction of ATF4 and its downstream target, CHOP, by promoting eIF2α dephosphorylation.	[14]
*Bornaviridae*	Borna disease virus (BDV)	+ (p, n)	ND	ATF4 nuclear localization increases in cerebellar cells but not in the hippocampus of infected animals.	[15]
*Caliciviridae*	Rabbit hemorrhagic disease virus (RHDV)	+ (t)	ND	*ATF4* and *CHOP* mRNA levels increase are associated with apoptosis induction.	[16]
*Circoviridae*	Porcine circovirus type 2 (PCV2)	+ (p)	[+]	The infection activates the PERK/eIF2α/ATF4/CHOP axis.	[17]
		+ (p)	ND	The viral proteins Replicase and Capsid induce the PERK/eIF2α/ATF4/CHOP axis.	[18]
		+ (t, p)	[+]	The viral protein ORF5 induces autophagy via the PERK/eIF2α/ATF4 and mTOR/ERK1/2/AMPK signaling pathways.	[19]
*Coronaviridae*	Coronavirus infectious bronchitis virus (IBV)	+ (p)	[+]	ATF4 is up-regulated through PERK- and PKR-mediated eIF2α phosphorylation.	[20]
	Nephropathogenic infectious bronchitis virus (NIBV)	+ (t, p)	ND	Upon infection, the BiP/PERK/ATF4 signaling pathway is activated and induction of renal apoptosis is observed.	[21]
	Porcine deltacoronavirus (PDCoV)	+ (t)	[−]	The infection activates the PERK/eIF2α/ATF4 axis and induces host translation attenuation.	[22]
	Porcine epidemic diarrhea virus (PEDV)	+ (t, p, n)	ND	The ATF4 protein is present in apoptotic cells.	[23]
	Severe acute respiratory syndrome coronavirus 2 (SARS-CoV-2)	− (p)	ND	Despite ISR activation and translational arrest, ATF4 and CHOP protein levels are not increased in infected cells.	[24]
*Flaviviridae*	Bovine viral diarrhea virus (BVDV)	+ (p) +/− (n)	[+]	Cytopathic BVDV induces ATF4 nuclear translocation and activates autophagy. Non-cytopathic BVDV induces ATF4 perinuclear localization but no autophagy.	[25]
	Dengue virus (DENV)	+ (n)	ND	None.	[26]
		+ (p)	[+]	None.	[27]
	Hepatitis C virus (HCV)	+ (p)	ND	ATF4 and ATF6 pathways contribute to the induction of CHOP in HCV replicon cells that showed an increased vulnerability to oxidant injury.	[28]
		+ (t)	ND	HCV induces chronic ER stress.	[29]
		+ (p)	ND	The viral core protein induces the PERK/ATF4 branch of the UPR which up-regulates the autophagy gene *ATG12*.	[30]
		+ (t)	ND	ATF4 may contribute to autophagy regulation during infection.	[31]
		+ (t, p)	ND	Cells expressing HCV proteins and exposed to oxidative stress adapt to cellular stress through eIF2α/ATF4 activation.	[32]
	Japanese encephalitis virus (JEV)	+ (t)	ND	None.	[33]
		+ (t)	ND	None.	[34]
		+ (p)	[−]	The viral protein NS4B activates PERK, which induces apoptosis via the PERK/ATF4/CHOP pathway.	[35]
	Tembusu virus (TMUV)	+ (t, p)	ND	CHOP induction leads to caspase-3 activation.	[36]
	West Nile virus (WNV)	+ (p, n)	[+]	ATF4 is involved in the up-regulation of GSH levels and the inhibition of stress granule formation induced by infection.	[37]
	Zika virus (ZIKV)	+ (t)	ND	Upon infection, ATF4 transcript level is weakly increased.	[38]
		− (t)	ND	None.	[34]
		+ (p)	ND	The infection transiently activates ATF4 but phosphorylation of PERK and eIF2α is sustained.	[39]
*Hepadnaviridae*	Hepatitis B virus (HBV)	+ (p)	ND	The reduction in intracellular ATP levels by the viral protein HBx induces ATF4 binding to the promoter of the *COX2* gene and its transcription.	[40]
		− (p)	ND	The viral HBx protein localizes in the ER lumen and directly interacts with BiP. This interaction results in suppression of eIF2α phosphorylation, which decreases the levels of ATF4/CHOP/Bcl-2.	[41]
		+ (t, n)	ND	HBV, with viral polymerase carrying the rt269L polymorphism, improves mitochondrial dynamics and enhances the autophagic flux, mainly thanks to the activation of the PERK/eIF2α/ATF4 signaling.	[42]
*Herpesviridae*	Epstein–Barr virus (EBV)	+ (p)	ND	LMP1 increases the ATF4 protein level through PERK/eIF2α phosphorylation. ATF4 transactivates *LMP1*.	[43]
	Human cytomegalovirus (HCMV)	+ (t,p)	ND	The infection activates PERK, but the amount of phosphorylated eIF2α is limited and no translation attenuation is detected.	[44]
		+ (p)	ND	The viral protein pUL38 induces phosphorylation of PERK and eIF2α, resulting in the accumulation of the ATF4 protein and cell protection against ER stress.	[45]
		+ (p)	ND	The viral protein UL148 activates ATF4 mainly through the PERK/eIF2α pathway.	[46]
	Human herpes virus 6A (HHV-6A)	+ (p)	ND	Induction of the PKR/eIF2α pathway results in a moderate increase in the ATF4 protein level, which peaks at the final stages of infection.	[47]
	Human herpes virus-8 (HHV-8)	+ (t, p)	[+]	ATF4 induces MCP-1 production and pro-angiogenic properties in endothelial cells.	[48]
		+ (p)	[+]	The viral protein ORF45 increases eIF2α phosphorylation and ATF4 translation, which in turn up-regulates the expression of lysosome-associated membrane protein 3 (LAMP3).	[49]
	Herpes simplex virus-1 (HSV-1)	+ (t, p)	ND	HSV-1 disarms the ER UPR in the early stages of viral infection. The activity of the eIF2α/ATF4 signaling increases at the final stage of HSV-1 replication.	[50]
	Murine cytomegalovirus (MCMV)	+ (p)	[+]	MCMV activates the PERK/ATF4 pathway but only induces a subset of ATF4 targets. ATF4 is required for efficient viral DNA synthesis and late gene expression during a low-multiplicity infection.	[51]
	Murine gamma herpes virus 68 (MHV68)	+ (p)	[−]	In response to ER stress, ATF4 inhibits B-cell receptor (BCR)-mediated MHV68 lytic gene expression by directly inhibiting the transcription of *RTA*, the MHV68 lytic switch transactivator. In a negative feedback loop, UPR-induced CHOP is required for and promotes BCR-mediated MHV68 lytic replication by decreasing upstream BiP and ATF4 protein levels.	[52]
	Pseudorabies virus (PRV)	+ (t)	[+]	The eIF2α/ATF4 pathway is activated during infection. PRV-induces apoptosis in later stages of infection through the CHOP/Bcl-2 axis. Overexpression of *BiP* or ER stress-inducing treatment can enhance PRV production.	[53]
		+ (t, p)	[+]	Infection-induced ER stress leads to PERK activation and up-regulation of ATF4, CHOP, and GADD34.	[54]
*Paramyxoviridae*	Newcastle disease virus (NDV)	+ (p, n)	[+]	The PKR/eIF2α/ATF4 pathway leads to an increase in the GADD34 protein level. GADD34, in conjunction with PP1, dephosphorylates eIF2α and restores global protein translation, benefiting virus protein synthesis.	[55]
		+ (p)	[+]	Induction of the PERK/eIF-2α/ATF4/CHOP signaling pathway is involved in the cyclin D1-dependent G0/G1 phase cell cycle arrest.	[56]
	Sendai Virus (SV)	+ (p)	ND	IRF7 up-regulates ATF4 activity and protein level, whereas ATF4 in return inhibits IRF7 activation.	[57]
*Parvoviridae*	Porcine parvovirus (PPV)	+ (t)	[−]	CHOP inhibits PPV replication by promoting apoptosis. ATF4 knockdown promotes PPV replication.	[58]
*Picornaviridae*	Foot-and-mouth disease virus (FMDV)	+ (p)	[+]	The capsid protein VP2 induces autophagy through the eIF2α/ATF4/AKT/mTOR cascade, and interacts with HSPB1.	[59]
	Group B coxsackievirus (CVB)	− (p)	[+]	PERK is activated and eIF2α is phosphorylated, but ATF4 protein levels do not increase. The ATF4/CHOP branch is blunted, thus inhibiting cell death.	[60]
*Poxviridae*	Myxoma virus (MYXV)	+ (t) − (p)	ND	PERK is activated and eIF2α is phosphorylated, but ATF4 translation is inhibited, which prevents *MCL1* and *CHOP* transactivation.	[61]
*Reoviridae*	Reovirus	+ (p)	[+]	The relative impact of ATF4 on viral replication depends on the infecting viral strain.	[62]
*Rhabdoviridae*	Vesicular stomatitis virus (VSV)	ND	[+]	None.	[57]
*Togaviridae*	Chikungunya virus (CHIKV)	− (t)	ND	ER UPR induction is primed since the phosphorylation of eIF2α and partial splicing of the *XBP1* mRNA are detected, but the viral protein nsP2 inhibits the transcription of a reporter gene under the control of the ATF4 promoter.	[63]
	Venezuelan equine encephalitis virus (VEEV)	+ (p)	ND	None.	[64]

(*) ATF4 activation induced by viral infection is assessed by the increase (+) or not (−) of ATF4 protein (p) or transcript (t) levels or by ATF4 nuclear localization (n). (**) The effect of ATF4 on viral replication is positive [+], negative [−] or not determined (ND).

**Table 2 biology-13-00146-t002:** ATF4 promotes HIV-1 replication and reactivation.

Models		Major Findings	Ref.
Replication	HIV-1 infected CD4^+^ Jurkat T cells.	Cell transfection with an ATF4-encoding plasmid up-regulates the HIV-1 proviral genome levels (qPCR of *gag* gene) and increases viral release (ELISA of p24).	[6]
	293 T cells transiently transfected with a plasmid encoding the HIV-1 genome and *GFP* gene.	siRNA directed against *ATF4* transcripts decreases the viral release (ELISA of p24) and Gag protein level (WB).	[7]
Reactivation	U1 cells *	Cell nucleofection with an ATF4-encoding plasmid increases the viral production in the cell culture supernatant (qPCR of *gag* gene and p24 levels by WB).	[6]
	J-Lat A1 ** and U1 cells * treated by a GCN2 inhibitor or supplemented with amino acids.	Inhibition of GCN2/ATF4 signaling represses the transcription of HIV-1 (real time qPCR with LTR primers).	[8]
	J-Lat cells and CD4^+^ T cells from HIV-1 infected individuals	FOXO1 inhibitor-induced reactivation of HIV-1 is reduced by pharmacological inhibition of PERK/ATF4 (*GFP* reporter gene or dddPCR with LTR primers).	[9]
	J-Lat A1, 2D10 *** cells and primary CD4^+^ T cells	Induction of the ISR/ATF4 signaling with a specific agonist of BiP, induces HIV-1 transcriptional activity (real time qPCR with LTR primers).	[10]

* U1 cell lines: subclone of the human U937 monocytic cell line that is infected with HIV-1. ** J-Lat A1 cell lines: Jurkat T cells containing one integrated copy of HIV-1 LTR that controls expression of the *EGFP* reporter gene. *** 2D10 cells: Jurkat T cells carrying a lentiviral vector that expresses (i) a mutated Tat carrying the H13L mutation, which enhances viral gene silencing, (ii) *Rev*, and (iii) a destabilized green fluorescent protein-encoding gene, which has a reduced half-life, in place of the *Nef* gene.

**Table 3 biology-13-00146-t003:** ATF4 target genes involved in apoptosis and HIV-1 infection.

ATF4 Target Genes	Model Related to HIV-1 Infection	Major Findings	Refs.
*BIM*/*BCL2L11*	T cells derived from *BIM*^−/−^ knockout mice treated with Tat.	BIM facilitates Tat-induced apoptosis.	[194]
	CD4^+^ T cells from pathogenic SIVmac251-infected rhesus macaques.	Infection by SIV up-regulates death ligand CD95L and proapoptotic BIM and BAK but not BAX protein levels.	[116]
	Latently HIV-1-infected macrophages and lymph nodes, and brain of HIV-1-infected individuals without detectable viral replication.	BIM is up-regulated and recruited into mitochondria both in vitro and in vivo in latently infected cells that are protected from apoptosis.	[195]
	SH-SY5Y cells treated with Tat.	FOXO3 down-regulates *BCL2* transcript and protein levels and up-regulates *BIM* transcript and protein levels after entering the nucleus, eventually causing cellular apoptosis.	[196]
	Monocytes-derived macrophages purified from PBMCs *.	Immunofluorescence analysis shows structural alterations in the mitochondrial architecture and an increase in BIM protein levels in the cytoplasm of infected cells.	[197]
*TID1/DNAJA3*	CEM-GFP cells transfected with a plasmid encoding the HIV-1 genome and *GFP* gene.	HIV-1 infection increases *TID1* transcript levels.	[198]
	HEK-293T cells transfected with either a Luciferase-encoding reporter vector or a plasmid encoding the HIV-1 genome and *GFP* gene.	TID1 increases HIV-1 LTR-driven gene expression and the viral p24 antigen release.	[199]
*G0S2*	PBMCs and myeloid monocyte-derived dendritic cells treated with virus-like particles containing the HIV-1 Pr55gag precursor protein and gp120 molecule anchored through the trans-membrane portion of the Epstein–Barr virus gp220/350.	*G0S2* transcript levels are increased in dendritic cells.	[200]
	THP-1 cells infected with a replication- deficient HIV-1 encoding the envelope glycoproteins from the vesicular stomatitis virus (VSV-G).	*G0S2* transcript levels are down-regulated in cells containing an integrated provirus, compared to bystander uninfected cells or cells harboring pre-integration viral complexes.	[201]
*MCL1*	PBMCs of HIV-1-infected individuals.	Apoptosis and viral load are inversely correlated with *MCL1* mRNA levels.	[202]
	Monocyte-derived macrophages purified from PBMCs.	The expression of the *MCL1* gene is up-regulated in macrophages infected with wild-type HIV-1 and in mock-infected macrophages that had been stimulated with M-CSF. However, *MCL1* is not up-regulated in macrophages infected with a *Δenv* HIV-1.	[203]
	PBMCs of HIV-1-infected patients before and during successful ART.	After 12 months of therapy, the expression of *MCL1* appears significantly up-regulated.	[204]
	Monocyte-derived macrophages or monocyte-derived dendritic cells incubated with R5 HIV-1 Bal.	HIV-1 infection decreases the Mcl-1 protein level but increases Bax and Bak.	[205]
	Vpr-treated monocyte-derived macrophages.	Resistance to Vpr-induced apoptosis is specifically mediated by *cIAP1/2* genes independently from Bcl-xL and Mcl-1, which play a key role in maintaining cell viability independently of the viral protein.	[206]
	HIV-1-infected macrophages and microglia.	Cells become viral reservoirs in response to acute infection through a BIM-dependent mechanism.	[195]
	THP-1-derived macrophages.	HIV-1 infection increases expression of the anti-apoptotic genes *MCL1*, *BCL2* and *BCL2L1* that encodes Bcl-xL.	[207]
	PBMCs of uninfected donors and HIV-1-positive patients treated by cART *.	Overexpression of *MCL1* is detected in PBMCs of cART-treated patients.	[208]
	Neutrophils from either healthy individuals, or HIV-1 patients whether asymptomatic, symptomatic, or ART receivers.	HIV-1 infection increases *MCL1* transcript levels in vivo, and ART partially reduces this increase.	[209]
*NOXA/PMAIP1*	Human CD4^+^ T cells infected with HIV-1 viruses lacking *Env*, *Vpr*, or *Nef*. Human PBMCs infected with wild-type HIV-1 viruses of different tropisms.	HIV-1 infection increases *NOXA* transcript levels, which is associated with cell death.	[210]
*PUMA/BBC3*	Circulating CD4^+^ lymphocytes from untreated HIV-1 infected donors.	HIV-1 infection increases Puma protein levels, which drop upon ART.	[211]
	HIV-1-associated encephalitis brain sections.	HIV-1 infection increases the Puma protein level in dying syncytia and neurons.	[212]
	Murine cortical neuron culture treated with gp120 III.	Gp120 III is sufficient to increase Puma protein levels and induce cell death.	[213]
	CD4^+^ primary T cells infected with HIV-1 lacking *Env*, *Vpr*, or *Nef* genes.	The *Env*, *Vpr* and *Nef* are not necessary for HIV-1-induced *PUMA* transcript levels increase and HIV-1-mediated cell death.	[210]
*TMBIM5* */GHITM*	Monocytes from control and HIV-1 patients.	*TMBIM5* transcript levels are decreased in HIV-1-infected monocytes.	[214]
	Brain from HIV-1-HAND * patients.	*TMBIM5* transcript levels are increased mainly in HIV-1-HAND patient astrocytes.	[215]
*TP53BP2/ASPP2*	Primary cortical neuron cultures treated with gp120 protein.	A high dose of gp120 stimulates the interaction of TP53BP2 with p53, which induces *BAX* transcription and contributes to caspase-3 cleavage.	[216]
	SH-SY5Y neuroblastoma cells treated with gp120 protein.	TP53BP2 regulates autophagy and apoptosis differently depending on the dose of gp120.	[217]

* PBMC: Peripheral blood mononuclear cell; cART: combined antiretroviral therapy; VLDL: very-low-density lipoprotein; HAND: HIV-1-associated neurocognitive disorder.

## Data Availability

Data sharing not applicable.

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
