# Peer review of "ATF4 Signaling in HIV-1 Infection: Viral Subversion of a Stress Response Transcription Factor"

_biology, 2024, doi:10.3390/biology13030146_

Round 1

Reviewer 1 Report

Comments and Suggestions for Authors

Comments for the Authors

biology-2858230

ATF4 signaling in HIV infection: viral subversion of a stress response transcription factor

Authors: Adrien Corne, Florine Adolphe, Jérôme Estaquier, Sébastien Gaumer, Jean-Marc Corsi

In this manuscript, Corne et al. review the role of a transcription factor, activating transcription factor 4 (ATF4) associated with the HIV-1 pathogenesis at different stages, including infection, replication, and latency. This review article emphasizes the mechanistic interplay between HIV-1 and cellular stress response, metabolism, and immunity. In addition, the interaction of the HIV-1 Vpu protein and the ATF4 protein is also covered in this manuscript. 

Overall, although the content of this manuscript is rich, additional topics related to HIV-1 virology and ATF4 are still worthy of a decent discussion (comments #5 and #6). The authors please find the suggestions below.

1. Both “HIV” and “HIV-1” are used in the manuscript and I do not see the reason the authors have to pick different terms for specific discrimination. I suggest that the authors choose either of them and keep the consistency throughout this manuscript.

2. Please also include HIV-1 in Table 1.

3. Line 151: the authors cite three references in this sentence. Thus I would suggest that the word “several” can be replaced by “different” or “three” groups.

4. Line 239: “Increase” and “decrease” viral replication do not sound familiar to me. I suggest the authors can choose the words “promote” and “suppress”.

5. Antiretroviral therapy and functional cure are of intense research interest for HIV-1 research, and a series of studies, focusing on antiretroviral drugs/therapies and ATF4, have been conducted (Brüning et al., 2013, Mol Oncol; Liu et al., 2015, PLoS One; Gassart et al., 2016, PNAS; Barbieri et al., 2018, J Cell Biochem; Ikebe et al., 2020, Blood Adv; Fraichard et al., 2021, Int Mol Sci; Subeha et al., 2021, Cancers; Mclntyre et al., 2023, Cell Rep). Please discuss this topic and include it in the manuscript.

6. HIV-1 infections can occur in the brain as well and cause neurocognitive dysfunction, namely HIV-associated neurocognitive disorders (HAND). Please also discuss the role of ATF4 associated with HIV-1 infections in the brain (Akay et al., 2012, Neuropathol Appl Neurobiol; Ma et al., 2016, Mol Neurobiol; Liao et al., 2016, J Neuroinflammation; Solomon et al., 2020, Mol Neurobiol).

Reviewer 2 Report

Comments and Suggestions for Authors

This review nicely summarizes the interplay between the stress response transcription factor ATF4 and HIV. Authors also highlight potential roles of ATF4 during HIV replication that have remained unexplored and deserve investigation.

There is no reviews on this topic available in the literature and thus, this manuscript is of interest and timely. My only suggestion for the authors is to revise and correct some speculations such as "Given the role of ATF4 in the regulation of TLRs, and that microbial antigens may activate innate cells, ATF4 activation could explain the inflammation that characterizes people living with HIV and therefore merits further clarification"This because it has been shown that chronic inflammation in PLWH is a multifactorial process involving at least viral replication, co-infections, microbial translocation, ART toxicity and loss of regulatory cells.

Round 2

Reviewer 1 Report

Comments and Suggestions for Authors

Thank you to the authors for carefully addressing each of my comments. The quality of this manuscript is now greatly enhanced and I do not have further comments.